# Leading from the bottom: The clinical leaders roles in an HIV primary care facility in Eldoret, Kenya

Felishana Cherop[1]*, Juddy Wachira[2], Vincent Bagire[3], Michael Korir[1]

1 Department of Management Science and Entrepreneurship, Moi University, Eldoret, Kenya, 2 Deparment of Mental Health and Behavioural Sciences, Moi University, Eldoret, Kenya, 3 Department of Business Administration, Makerere University Business School, Kampala, Uganda

☯ These authors contributed equally to this work.
* fcherop@gmail.com

## Abstract

### Background

Clinical leaders in health systems play critical roles in making decisions that impact patient care and health system performance. Current literature has focused on the importance of clinical leaders' roles in healthcare settings and has not addressed the leadership aspect that clinical leaders engage in day-to-day decision-making in HIV facilities while providing HIV patient care. Therefore, identifying the leadership roles that wclinical leaders perform at HIV primary facilities is of critical importance.

### Purpose

The study explored the views of healthcare providers working in AMPATH-MTRH HIV facility on what they perceived as the roles of clinical leaders at the HIV primary care facility.

### Methods

We conducted a qualitative exploratory study between December 2019 to May 2020, involving in-depth interviews with (n = 22) healthcare providers working in AMPATH-MTRH HIV facility, who were purposively and conveniently sampled to participate in in-depth interviews to explore perceptions regarding the leadership roles of clinical leaders. The collected data were analyzed thematically and Nvivo vs.12 software was used for data management.

### Results

The following themes were identified from the analysis regarding perceived clinical leaders' roles in an HIV primary care facility: 1) **Strategic roles:** providing direction and guidance, ensuring goals and objectives of the department are achieved within the set timelines, planning, and budgeting for adequate resources to support patient HIV care **2)** Interconnecting health systems levels and supervisory oversight roles: a link between management, staff, and patients, solving problems, organizing and attending departmental meetings, facilitate staff training, accountable, collaborating with other departments and leaders, defines and

**Data Availability Statement:** All relevant data are within the paper and its Supporting Information files.

**Funding:** "This research (or "[FC]") was supported by the Consortium for Advanced Research Training in Africa (CARTA). CARTA is jointly led by the African Population and Health Research Center and the University of the Witwatersrand and funded by the Carnegie Corporation of New York (Grant No. G-19-57145), Sida (Grant No:54100113), Uppsala Monitoring Center, Norwegian Agency for Development Cooperation (Norad), and by the Wellcome Trust [reference no. 107768/Z/15/Z] and the UK Foreign, Commonwealth & Development Office, with support from the Developing Excellence in Leadership, Training, and Science in Africa (DELTAS Africa) programme. The statements made and views expressed are solely the responsibility of the Fellow. For the purpose of open access, the author has applied a CC BY public copyright licence to any Author Accepted Manuscript version arising from this submission." The funders had no role in study design, data collection, and analysis, decision to publish or preparation of the manuscript.

**Competing interests:** The authors have declared that no competing interests exist

assigns responsibilities, ensure quality patient service, coordination, and management of daily activities **3) Research roles**: data collation, analysis, generation, review and reporting to the management.

## Conclusion

Clinical leaders in the HIV care system perform leadership roles that are characterized by strategic, middle-level, supervisorial and research which reflects the model of the leadership and management style of the HIV care system. The understanding of these roles contributes valuable insights to HIV leaders and managers to recognize the important contribution of clinical leaders and consider reviewing Standard Operating Procedures to include these leadership roles and strengthen their capacity to maximize clinicians' contribution to improve HIV care and enhance responsive health systems.

## Background

Clinical leadership is recognized as an important element in improving healthcare services and strengthening health systems, however, it has received little attention in low and medium-income countries (LMICs) [1]. Clinical leadership has been defined as one that takes place in any clinical setting whose aim is to enhance care and positive patient outcomes through the provision of quality patient service when leading staff in a dynamic environment that has multifaceted client composition [2]. It has been considered the strongest precondition, and clear leadership that promotes integrative and proactive care through the facilitation of interdisciplinary collaboration [3].

Today, clinical leaders are considered the champions of leading change and improving service in their organizations [4,5]. Their greater involvement in leadership hierarchies such as strategic leadership positions improves the quality of strategic decisions and successful implementation [6,7]. They act as resource persons, preceptors, mentors/coaches and role models in demonstrating critical and reflective thinking as well as establishing and monitoring standards of practice to improve patient care [8]. In Ireland, they were perceived to facilitate evidence-based practice in care delivery, mentor students and co-workers, and, utilize clarity in decision-making [8,9]. However, evidence of mentorship of health personnel to improve the quality of health care in low and middle-income countries remains elusive [10]. From a nursing perspective, clinical leaders influenced care delivery systems through engagement in policy development at all levels of government [1,11], but these perspectives are less documented in the HIV context. Similarly, there is confusion in clinical leadership roles in distinguishing between managerial, resource management, and clinical roles [12].

Additional roles of clinical leaders include setting direction, providing vision and promoting professionalism, promoting interprofessional collaborations, and having the resources to perform tasks effectively [13]. They establish a collaborative atmosphere, structuring work to ensure patients get the best nursing care, customize their presence in the practical work with patients according to standard operating guidelines, and monitor co-workers' professional practice [14]. However, these roles have not been categorized according to different hierarchies of leadership in most health facilities and unclear whether clinical leaders in similar or different settings such as HIV performed the same leadership roles. In Kenya for instance, the use of distributed leadership to examine clinical leadership in hospitals was useful in analyzing middle-level leadership [1], however, it is unknown if this type of leadership is effective among

clinical leaders in HIV primary care. In Sub-Saharan Africa, supportive supervision increased job satisfaction, and health worker motivation and improved and maintained crucial primary healthcare quality standards [15–17]. However, there is mixed evidence on whether it would translate to increased clinical competency and effect on clinical outcomes [16]. In South Sudan, supportive leadership remained a daunting task in the health sector due to a combination of external and health system factors for instance, many supervisors had no formal training on supportive supervision and action plans developed during supervision were inadequately followed up due to insufficient funding [18].

Whereas clinical leadership is important at the operational level in making critical clinical decisions daily [19], there is a paucity of clinical leader roles in HIV primary care to steer quality HIV patient care and improve health system performance. Our study explored the perspectives of healthcare providers on the leadership roles of clinical leaders in an HIV primary care facility in Eldoret, Kenya.

## Methods

### Study design

A qualitative exploratory study was conducted between December 2019 to May 2020 involving in-depth interviews to explore the views of healthcare providers to obtain an in-depth understanding of the perceived leadership roles of clinical leaders that may influence patient outcomes and health system improvements.

### Study setting

The study was conducted in the Academic Model Providing Access to Healthcare- Moi Teaching and Referral Hospital (AMPATH-MTRH). AMPATH is a consortium of different institutions that provide comprehensive HIV care to the population in western Kenya. It supports Ministry of Health (MoH) facilities in more than 15 counties in Western Kenya and is organized into cluster clinics that serve adult patients, children, and, adolescents between 8 am to 5 pm from Monday to Friday. The facility has a systematic structure of clinical management that guides patient care (S1 Fig). Each of the clinics in MTRH-AMPATH is headed by a clinical leader who performs both leadership and clinical functions [20,21]. The providers received instructions from the clinical leader and they interacted daily in providing patient care. Therefore, they would be better positioned to provide perspectives on what they think are the leadership roles performed by clinical leaders.

### Participant sampling and recruitment

The participants in our study involved frontline healthcare providers (clinicians, nurses, counselors, and a pharmacist). The total population of healthcare providers at MTRH-AMPATH was 50 at the time of the study. We first presented the research permits for conducting the study in AMPATH to the clinical leaders in charge of various clinics to explain the purpose of the study and obtain approval to interview the HCPs. We purposively and conveniently approached 25 HCPs individually and 22 consented to participate in in-depth interviews. Literature suggests that a sample size of 10 is adequate if a homogenous group of people is interviewed, nevertheless, a sample size of 15–30 is considered appropriate for a qualitative study [22].

**Data collection procedures.** Data collection was done using semi-structured interviews which were conducted in a private room that was identified for the study to protect confidentiality and give ample time for the participant to process information. The interviews lasted

between 40–60 minutes upon consent. After interviewing 22 participants, the authors felt that data saturation had been achieved [23,24] A semi-structured interview guide was used to guide the in-depth interviews with a set of questions that sought participants' insights, particularly to describe the leadership roles of clinical leaders in an HIV primary care facility and included demographic information such as expertise, sex, and age. The views of the study sample reflected the perceptions of the providers in the HIV facility. Also, the audio-recorded data were stored in a safe digital format that was later used for data transcription, interpretation, and analysis.

## Data management and analysis

Thematic analysis was used to analyze qualitative data [23,25]. The first author began by transcribing the audio recordings verbatim then the transcripts were read and re-read by all the authors to familiarize themselves with the content of the data and understand its richness and diversity. The transcribed data was imported to NVivo12 for management and coding. Two authors with qualitative backgrounds coded the data, compared, discussed, and examined the codes and categories to identify patterns and themes for interpretation. All the authors then read and discussed the themes and agreed on the final themes to be included in the report. A final codebook (S1 Table) indicating emerging themes was then developed that informed the final write-up of the results and supported by relevant quotes from the data. To ensure validity and reliability, and increase the quality of the data, the codes were counterchecked and appraised by all authors to confirm there was no repeated and conflicting content [26,27]. The data collection procedures and analysis were made clear to help maintain consistency and rigor throughout the study. Team debrief sessions were held during the data collection to enhance the quality of the data. The analysis informed descriptive categories/domains that represent thematic healthcare providers' perceptions of the clinical leadership roles of clinical leaders in an HIV primary care facility.

## Ethical considerations

This study was granted ethical approval by the Institutional Research Ethics Committee (IREC) in Moi Teaching and Referral Hospital (MTRH); (Approval No.0003485), and a research license from the National Commission for Science, Technology, and Innovation (NACOSTI No NACOSTI/P/20/3253) before data collection. Participation in the study was voluntary and the healthcare providers signed written informed consent.

## Results

### Participant demographic characteristics

Twenty-two (88%) out of twenty-five healthcare providers (HCPs) participated in the in-depth interviews upon data saturation. Most of the participants were clinical officers 14(63.6%), followed by nurses 5(22.8%), a few were counselors 2(9.1%) and a pharmacist (4.5%). There were more males 12(54.5%) than females 10(45.4%), and all 22 (88%) earned a monthly income of (>500$) and 22 (88%) had over 1 year of experience in providing HIV care and had an average age of 41–50 years.

### Domains of clinical leader roles

Participants described three themes that characterize clinical leadership roles in an HIV primary care facility: strategic leadership roles, interconnecting health systems levels, and supervisory oversight roles and research roles.

**Strategic leadership roles.** Clinical leaders in the HIV care system act as team leaders in providing direction and guidance to staff and patients by informing and guiding them in understanding the program, its objectives, strategies, and visions. This is done by informing staff and patients of their expectations and setting targets to be achieved. This would ensure that all the stakeholders are involved in the process to ensure the goals and objectives of providing quality services to patients are prioritized and achieved.

"So, according to me leadership is like giving the way forward for the people who are working and you are leading them to where you are supposed to go. So, you give the way to the people who are under you so that they know their objectives and how to achieve their objectives" (Participant #1, Clinical Officer)

"I believe a leader is someone who takes responsibility for all actions and sets targets that are given by your program and that person shows it by example" (Participant #2, Pharmacist)

Clinical leaders ensure that the objectives of the department are achieved by making sure that all the departments are meeting their targets or the clients are served without complaint so that the clients are satisfied with our services.

"You know, in every organization, you must have what we call objectives or goals. So, as a leader, you must work so that you achieve those goals and objectives. When you want to achieve those goals, you have to make sure that everybody is involved. Like a leader, if you have a task, you identify an individual who can do that task and it can work" (Participant #3, Clinical Officer)

Providers described that clinical leaders carry out planning and budgeting for prioritization and allocation of limited resources particularly in a large care system that has high patient volumes, to ensure a continuous provision of strategic services to patients and strengthening of the health system's responsiveness. This includes enough supplies of commodities to avoid stockouts when needed, adequate staffing, organizing work shifts to ensure present staff at the facility even during holidays, availability of medical supplies, and finance.

"Another key thing that my leader is doing is the planning and budgeting for the allocation of resources, the allocation of staff such that who is allocated to work where and at what time. When there are emergencies, he assigns who is going to intervene, what are the challenges, and who is going to handle them" (Participant #4Nurse)

**Interconnecting health systems and supervisory oversight roles.** Participants reported that clinical leaders play an important role in interconnecting different levels of health systems management by facilitating communications, collaboration, and coordination among various stakeholders. Through effective supervisory oversight, the leaders ensure the delivery of high-quality HIV care while fostering professional growth and development within their teams.

(a)Interconnecting health system levels

Participants recognized that a clinical leader acts as a link between different levels of management, patients, and frontline staff. They would coordinate and facilitate communication channels, relay feedback from staff and patients to management, and ensure that the facility goals align with frontline needs such as linking the patients with other care providers including consultants and specialists to ensure that patients receive specialized care

"We have our in charge and most of the time she is the one who connects us with the management level, if we have issues, she is the one to take the issues upwards, and if there is anything that has to be communicated again from the chief of party or the clinical manager, she is the one to relay the information" (Participant #5, Clinical Officer)

"I would say majorly it is coordination; he coordinates the activities and he is the link between the higher management and the client. He coordinates all the activities and the services within the department" (Participant #4, Nurse)

The healthcare providers identified problem-solving as a key role of clinical leaders in addressing technical issues related to patient care, staff conflicts, and challenges in the working environment, such as complaints from the patients, difficult cases that the junior staff would be unable to solve by providing counseling services to the patients, and having a session with the staff to discuss the strategies for addressing the challenges because it helps the clinic run effectively.

"And some patients have complaints; the delays, I was not treated well, somebody is handled badly. A leader should come down to earth to settle down the issues. Sometimes you apologize, one may have said something which is not good and hurts. A leader should come down to be able to apologize on behalf of that staff and clear the matter" (Participant #5, Nurse)

Participants described collaboration within and among the departments in a healthcare system as an important aspect of clinical leadership. Leaders in all departments would not work in isolation but would work in collaboration to borrow information from each other in terms of best practices and services. The leaders would normally interact during inter-departmental meetings and training. The care departments included the nursing, laboratory, clinical, records, nutrition, social work, guidance and counseling, and pharmacy while the support departments could include the finance, accounts, human resource, and legal office department. A client would be referred and receive services from various departments.

"Leadership is not limited to a specific department because in healthcare, it is a system and it is made up of several arms. Because it is made up of several arms, working together is inevitable, they have to reach out to one another "(Participant #8, Nurse)

"The healthcare system is intertwined; the departments being intertwined. And you realize that one department cannot function on its own, you really have to do some coordination and consultations with other departments" (Participant #9, Clinical Officer)

### (b) Supervisory Oversight

Providers described the clinical leader's role as defining and assigning roles and responsibilities to staff in their respective departments, which is a typical function of management where leaders irrespective of their levels must allocate tasks and resources to achieve them. They ensure that each team member understands their tasks and responsibilities and this would foster clarity and accountability.

"Number one is to do supervisory roles like daily supervision and to make sure people are at work and doing the right things and delivering. You know, implementing what they are supposed to be doing" (Participant #9, Clinical Officer)

Other participants described that the clinical leader would be accountable by making monitoring and follow-ups continuously to ensure the staff perform their duties diligently in a conducive work environment that promotes patient-provider relationships, provides timely reports, and represents the unit in all forums.

"So, being a leader, the major role which our leader plays to the care workers is to check on how we run our activities, whether we perform as healthcare providers, especially giving services to the patient. That is the major role that our group leader does" (Participant #7, Clinical Officer)

"Another thing is a representation of that unit because, in as much as you are working as a team at that place, this leader at some point will have to be accountable for the area in which they are working so that in case of any issues, you don't always have to fault others. So, this particular leader is the one who will carry the burden of being accountable and being responsible for that particular area" (Participant #9, Clinical Officer)

To ensure quality services to the patient, the clinical leader would create a conducive care environment that promotes quality service delivery and patient-provider relationships. The leader oversees day-to-day operations, intervenes in emergencies, manages data and reporting, and ensures that services are delivered on time and according to established standards. For instance, the laboratory should be equipped and functioning, obtaining feedback from clients for quality improvement of HIV services.

"A leader in a healthcare system ensures that our clients are given quality service. For example, a leader ensures that his team or her team are on duty and are timely with teamwork, so in the long run, we give quality service to our clients" (Participant #10, Clinical Officer)

Participants discussed that a clinical leader would organize and attend regular departmental meetings to discuss the program and departmental activities to provide progress reports. The meetings could range from daily, weekly to monthly and the clinical leader is usually the chair.

"We also have small brief meetings in the course day. We look at anybody who had a problem and what are the successes of the day. We look at all that. So, I think having regular meetings should be one of the things that a leader should inculcate in their team" (Participant #2, Pharmacist)

"He also organizes routine meetings; routine meetings at the clinic level with the staff to evaluate the data, to also check for matters arising, any issues" (Participant #6 #Clinical Officer)

Clinical leaders would serve as mentors to junior staff and guide them in implementing new policies, practice guidelines, and regulations such as the Standard Operating Procedures (SOPs).

"And then secondly, there are government policies like when they are rolling them down, she is the lead person like she mentors us" (Participant #11, Clinical Officer)

The clinical leaders would facilitate staff training and development initiatives through short courses, seminars, and workshops that would enhance skills development and ensure team members remain updated on best practices and emerging trends in healthcare delivery during the monthly Continuous Medical Education (CME).

"It could be we need to have the training and he is the one who arranges. Number three is representing us in activities outside here maybe acquiring some skills outside there and then coming back and training us. I think those are the most important ones that I see in our setup. You understand whereby care providers, all of them, cannot go for training. For example, it is on gender-based violence, he can represent us then at the end of the day, come and brief us on what has been trained" (Participant #7, Clinical Officer)

**Research roles.** Clinical leaders were described as researchers who would be responsible for the collation of data, analysis of data, generation, review, and submission of progress reports to the high levels of management. These may include, measurements and indicators of services provided to the clients and data review.

"Another role he does is to collect data; to collect specific data about the healthcare system and targets. He also gets the data analyzed and prepares reports. You realize that with HIV, there is so much data; those in care, those who have defaulted, those to follow-up, and all that" (Participant #6, Clinical Officer)

## Discussion

In our study, we explored the clinical leadership roles in an HIV primary care facility. To our knowledge, we believe this is the first study to describe these roles in the HIV context in Kenya. The roles of clinical leaders were categorized into three domains: strategic role, inter-connecting health systems levels and supervisory oversight, and research role.

A strategic leadership role was identified as key for clinical leaders in an HIV primary facility. Clinical leaders act as team leaders in providing guidance and direction on what is expected of the patients and providers. This suggested that they should understand the structure of the HIV system in terms of its goals, objectives, and strategies to translate them into actionable items and support other stakeholders in realizing them. Consistent with previous literature, a clinical leader was seen as being team-focused while maintaining relationships in and outside the team for effective decision-making and a key player in communication for their local area of practice [28]. Other studies have acknowledged the importance of visionary leadership [29–31], teamwork [32,33], treating team members with respect, and facilitating a conducive work environment [34,35] may enhance teamwork. In addition, the clinical leader in primary care should work strategically and with a vision for better practice across teams [2].

Our study emphasizes the importance of planning and budgeting by clinical leaders by prioritizing available resources highlighting the importance of strategic management knowledge to do forecasting activities of a department. Consistent with our findings, leaders should be strategic planners and, participate in policymaking [36]. However, the findings differ in the type of planning from that of previous literature which required clinical leaders to help patients in developing plans for achieving their treatment goals and setting directions [30,31]. Although clinicians would need to have a broad overview of a budget process, they were not expected to be accountants and should seek help for some financial tasks to manage a budget in a way that will reduce inefficiencies and provide the greatest benefit for patients [37].

Our study also indicates that clinical leaders interconnect different levels of management for example to represent patients' and providers' issues to the facility management and link patients with other specialists in the facility. They would do this to address problems that may exist within the health system using clear channels of communication. This aligns with

previous literature to solve problems such as patient complaints and other technical challenges [30,35,38–42]. Recent literature echoes similar findings that the role of clinical leaders in workforce development is to be responsive to direct care staff when they communicate their concerns. The leader encourages them to reach out proactively whenever they have concerns to be addressed [43].

The study also highlighted that clinical leaders should collaborate with other leaders in different departments to bring collective efforts and expertise in enhancing HIV patient care and health system performance. This emphasizes the need for dialogue and instilling confidence among co-workers. A similar perspective in the literature acknowledges that clinical leaders at the point of service would rely on their communication, collaboration, and coordination skills to motivate others to act because good communication was seen as the foundation for the effective coordination of activities [44]. Similarly, a staff nurse clinical leader who is at the bedside would establish a good atmosphere for collaboration through mutual respect, courage to be honest, and encouraging reflection [14].

In our study, clinical leaders performed supervisory duties by defining and assigning roles and responsibilities to staff in their respective departments which is contrary to a previous finding where departmental heads at the mid-level of management in healthcare institutions in Kenya were expected to tell clinical staff what to do, demonstrating a top-down approach to ensure formal responsibilities were met and accountability done within the departments [45].

The study provides an additional role for a clinical leader to be accountable by managing available resources to ensure quality patient care. Whereas this may be a strategic function of management, there is insufficient literature on the direct accountability of clinical leaders at microsystems given that their roles are clinically oriented. In previous literature, clinical leaders were not only perceived to improve and promote clinical excellence but were also expected to be professionally accountable, enhance multidisciplinary teamwork, and patient safety, and achieve greater value for money [46].

We found a general pattern that clinical leaders ensure quality services to the patient by creating a conducive working environment for the clients and the care providers. The leader would ensure that the patients received all the required services timely, professionally, ethically, and in a satisfactory manner. A related observation from previous studies in nursing leadership established that nurse leaders would consciously structure work to ensure patients' best possible nursing care [14], while the role of clinical leaders is to enhance quality and transform clinical services for excellence [2,47].

Our study points out that clinical leaders organize and attend regular meetings to discuss departmental performance that may affect patient care and identify areas that would require skills development of staff through training. In contrast, literature found limited opportunities for clinical and nurse leaders that would limit them in executing tasks in the same environment and discussing joint departmental issues even where the standard operating procedures demanded [1].

Clinical leaders were perceived to coordinate activities in the HIV facility to create harmony between and within departments and share knowledge and other resources. This was highlighted by previous literature as important for sharing ideas and best practices for providing quality HIV care [14,36]. However, poor coordination of work across cadres in primary healthcare delivery was highlighted by public health nurses [9], and the inability of some trained nurses to handle the supervision of other people [34].

Participants highlighted that clinical leaders would act as mentors to other healthcare providers and fellow leaders to pursue career development for example through continuous medical education that would enhance health system improvement through implementing new policies and guidelines. The findings concur with previous studies that physician leaders value

the role of mentorship because they believe that they are natural mentors, and they would have reached their current position because of someone who mentored them [48]. Also, clinical leaders are role models in leading by example, walking the walk, and not just talking particularly in dealing with day-to-day clinical presentations [49]. On the contrary, a study found that some nurse participants perceived their clinical leader as unwilling to share expertise, knowledge, and skills [28].

Clinical leaders were perceived to perform research roles through the collection of data, analysis, and providing progress reports from their departments. Although the clinical leaders may not be equipped with research scientific knowledge, additional training and involvement in the research process and output relating to HIV care would be necessary. This finding aligns with previous literature that clinical experts/leaders would link theory and practice and would encourage research and dissemination of knowledge [28]. They would initiate, conduct, and disseminate findings of locally based research in specialty and would be involved in larger research studies in Australia [50]. A study acknowledged the attributes of clinical leadership within a framework of quality that a clinical leader should have a vibrant, research-based, evidence-based practice culture [2]. However, a systematic literature review found that the focus on clinical leadership as a research target concerning integrated care appears a new phenomenon [51], demanding further investigation.

## Strength and limitation

To our knowledge, there has been little information published on the role of clinical leaders in HIV primary care facilities. Hence, the findings of this study would provide rich views and experiences of providers that shed more light on the varied roles of clinical leaders at the primary care facility level. The findings also provide important aspects for decision-making on quality improvement of service delivery in the HIV primary care setting. While we believe this is a novel study in the HIV context, some limitations are important to acknowledge. First, clinical leaders and patients were not interviewed who could have provided different perspectives on clinical leader roles and this demands a study to obtain their views to gain an understanding of issues surrounding being a clinical leader in an HIV care system. Secondly, the providers who were interviewed included clinical officers, nurses, counselors and a pharmacist, hence little representation of the range of experiences of other providers who were not captured in the study. Third, providers expressed their views and experiences concerning the perceived clinical leader roles in the present HIV facility, hence may have varied degrees of transferability in other settings with varied aspects such as geographical location and service delivery. Fourth, this study only focused on clinical leaders at the lower level of the organization, and may be interesting to obtain views from other leaders in the hierarchy of the organization (meso and strategic levels). Fifth, there is a danger of social desirability bias since most of the statements made by the healthcare providers tended to be positive and this may not reflect a true representation of their views and behaviour because they were evaluating their clinical leader in charge. Also, the set of guiding questions did not focus on relationship dynamics that relate to trust and hierarchies within the care system.

## Conclusion

This study explored the roles of clinical leaders in an HIV primary care facility. The leadership roles that were associated with clinical leaders included; strategic, middle-level, supervisory functions, and engagement in research. The understanding of these leadership roles performed by clinical leaders contributes valuable insights to health care leadership discourse, acknowledging the important contribution of clinical leaders to the HIV care systems and providing a

basis for enhancing clinical leadership within clinical settings such as HIV. It would be necessary to strengthen the capacities of clinical leaders to maximize their contribution to improve HIV care and enhance responsive health systems.

## Supporting information

**S1 Fig. AMPATH-MTRH clinical management flow chart.**
(DOCX)

**S1 Table. Codebook with minimal data.**
(DOCX)

## Acknowledgments

We sincerely appreciate the MTRH-AMPATH for allowing this study to be conducted. We also thank the healthcare providers for participating in this study.

## Author Contributions

**Conceptualization:** Felishana Cherop, Juddy Wachira, Vincent Bagire, Michael Korir.

**Formal analysis:** Felishana Cherop.

**Funding acquisition:** Felishana Cherop.

**Investigation:** Felishana Cherop.

**Methodology:** Felishana Cherop, Juddy Wachira, Vincent Bagire, Michael Korir.

**Supervision:** Juddy Wachira, Vincent Bagire, Michael Korir.

**Writing – original draft:** Felishana Cherop.

**Writing – review & editing:** Felishana Cherop, Juddy Wachira, Vincent Bagire, Michael Korir.

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
