## [Decision Letter · Decision Letter 0]

31 May 2023

PONE-D-23-05699Leading from the bottom: The strategic leadership roles of clinical leaders in an HIV primary care facility in Eldoret, KenyaPLOS ONE

Dear Dr. Cherop,

Thank you for submitting your manuscript to PLOS ONE. After careful consideration, we feel that it has merit but does not fully meet PLOS ONE’s publication criteria as it currently stands. Therefore, we invite you to submit a revised version of the manuscript that addresses the points raised during the review process. Particularly paying attention to the methodological issues.

We look forward to receiving your revised manuscript.

Kind regards,

Edward Nicol, PhD

Academic Editor

PLOS ONE

Journal Requirements:

"This research (or “[FC]”) was supported by the Consortium for Advanced Research Training in Africa (CARTA). CARTA is jointly led by the African Population and Health Research Center and the University of the Witwatersrand and funded by the Carnegie Corporation of New York (Grant No. G-19-57145), Sida (Grant No:54100113), Uppsala Monitoring Center, Norwegian Agency for Development Cooperation (Norad), and by the Wellcome Trust [reference no. 107768/Z/15/Z] and the UK Foreign, Commonwealth & Development Office, with support from the Developing Excellence in Leadership, Training, and Science in Africa (DELTAS Africa) programme. The statements made and views expressed are solely the responsibility of the Fellow. For the purpose of open access, the author has applied a CC BY public copyright licence to any Author Accepted Manuscript version arising from this submission."

4. Please ensure that you include a title page within your main document. You should list all authors and all affiliations as per our author instructions and clearly indicate the corresponding author.

5. Please ensure that you refer to Figure 1 in your text as, if accepted, production will need this reference to link the reader to the figure.

Reviewers' comments:

Reviewer's Responses to Questions

**Comments to the Author**

1. Is the manuscript technically sound, and do the data support the conclusions?

Reviewer #1: Yes

Reviewer #2: Partly

2. Has the statistical analysis been performed appropriately and rigorously? 

Reviewer #1: N/A

Reviewer #2: N/A

3. Have the authors made all data underlying the findings in their manuscript fully available?

Reviewer #1: No

Reviewer #2: Yes

4. Is the manuscript presented in an intelligible fashion and written in standard English?

Reviewer #1: Yes

Reviewer #2: Yes

5. Review Comments to the Author

Reviewer #1: Thank you for the opportunity to review this manuscript. I find the manuscript of relevance to the field. However, it requires some major revisions, so as to meet the ideal standard for publication. Details of the required revision are attached.

Reviewer #2: “Strategic” is overused and used in different in contexts. As this is a key term, please be more precise in what is encapsulated in “strategic.” This definition comes out slightly in the results but feels a but tautological to state that theme for perceived clinical leader’s strategic roles was being strategic (lines 19-20)

I am surprised the issue of hierarchy and trust (as well as other social dynamics) did not come more prominently in the findings and themes, and perhaps this reflects also potential social desirability bias or some considerations on reflexivity on what health workers are willing to disclose with regard to their “clinical leaders” Moreover, the findings/themes seem very basic and what you would expect to find on a terms of reference. There should be a re-analysis of themes to identify more profound findings that can add to the literature.

It is also not as clear as how the themes of strategic, managerial and supervisory are distinct themes. The themes should be better fleshed out to describe what strategic means and its facets.

It would also be interesting to overlay the background with literature on “supportive supervision” and “in-service training/mentorship.” This comes out in the discussion, but it’s not clear in the background how this is different and what value strategic leadership means over existing forms support.

In line 76-77, there is not really context for who respondents were acknowledging that it is a grey area.

The strengths and limitations should also include the potential biases, including social desirability bias, especially since most of the quotes and what was stated tended to be positive. Acknowledging that health workers are providing this feedback about their supervisors in the context of their work could put some limitations on the extent of information that would be shared.

6. PLOS authors have the option to publish the peer review history of their article (what does this mean?). If published, this will include your full peer review and any attached files.

Reviewer #1: No

Reviewer #2: No

---

## [Author Response · Author response to Decision Letter 0]

9 Jul 2023

I have uploaded a rebuttal letter providing response to reviewers' comments

---

## [Decision Letter · Decision Letter 1]

30 Aug 2023

PONE-D-23-05699R1Leading from the Bottom: The Clinical Leaders Roles in an HIV Primary Care Facility in Eldoret, KenyaPLOS ONE

Dear Dr. Cherop,

Thank you for submitting your manuscript to PLOS ONE. After careful consideration, we feel that it has merit but does not fully meet PLOS ONE’s publication criteria as it currently stands. Therefore, we invite you to submit a revised version of the manuscript that addresses the points raised during the review process.

The authors have not fully addressed comments raised earlier by both reviewers. These are needed to improve the quality of the manuscript. Kindly see below and attached documents for detailed comments. 

We look forward to receiving your revised manuscript.

Kind regards,

Edward Nicol, PhD

Academic Editor

PLOS ONE

Reviewers' comments:

Reviewer's Responses to Questions

**Comments to the Author**

1. If the authors have adequately addressed your comments raised in a previous round of review and you feel that this manuscript is now acceptable for publication, you may indicate that here to bypass the “Comments to the Author” section, enter your conflict of interest statement in the “Confidential to Editor” section, and submit your "Accept" recommendation.

Reviewer #1: (No Response)

Reviewer #2: (No Response)

2. Is the manuscript technically sound, and do the data support the conclusions?

Reviewer #1: Yes

Reviewer #2: Yes

3. Has the statistical analysis been performed appropriately and rigorously? 

Reviewer #1: N/A

Reviewer #2: Yes

4. Have the authors made all data underlying the findings in their manuscript fully available?

Reviewer #1: (No Response)

Reviewer #2: Yes

5. Is the manuscript presented in an intelligible fashion and written in standard English?

Reviewer #1: (No Response)

Reviewer #2: Yes

6. Review Comments to the Author

Reviewer #1: Thank you for the opportunity to review this submission once again.

Unfortunately, the authors have not fully addressed the comments raised earlier; and need to do so, to improve the quality of the manuscript.

Detailed reviewer comments are attached.

Reviewer #2: The abstract background gives a sense of over-using the term "strategic" and if this can be revised to find alternative appropriate wording.

The results tends to provide a lot of quotes back to back and would be good to see if there could be further analysis and quotes used more effectively. It otherwise just feels like excerpts from the transcripts within contextualizing paragraphs in between. This may perhaps mean having additional themes to improve readability as some themes run across several pages and are hard to process all the information. For example, lines 170-249 span several quotes in between them. I also note that within the theme of "managerial duties" it states (line 220) "A strong theme that emerged among the participants was the collaboration within and among the departments in a health care system." Should this therefore be a separate theme or rephrased.

7. PLOS authors have the option to publish the peer review history of their article (what does this mean?). If published, this will include your full peer review and any attached files.

Reviewer #1: No

Reviewer #2: No

---

## [Author Response · Author response to Decision Letter 1]

15 Jan 2024

Dear Reviewer, 

Thank you for the valuable comments you have provided in our submitted manuscript. We have responded to the comments provided for by Reviewer (R1), which were sent back to me. We have attached a rebuttal letter outlining the concerns and the response and pasted below this response. Thank you 

Rebuttal Letter Addressing Reviewer’s Comments

Reviewer #1

Second Review January 2024

Reviewer’s comments (R1)

1. Comment 2, made during the first review has not been addressed. “The results identified4 themes” is not grammatically right. The phrase therefore needs to be revised. 

Response: We have addressed this comment in the abstract and results sections. 

In the abstract: “The following themes were identified from the analysis regarding perceived clinical leaders’ roles in an HIV primary care facility: 1) Research roles….

In the results section: “Participants described four themes that characterize clinical leadership roles in an HIV primary care facility: strategic leadership roles, middle-management leadership roles, supervisory leadership roles and research roles”

2. Another comment asked for the purpose of the study/objectives to be included. This hasn’t been addressed. Instead, the authors have added literature, making the background section longer.

Response: 

At the end of the background, we have included “Our study explored the perspectives of healthcare providers on the leadership roles of clinical leaders in an HIV primary care facility in Eldoret, Kenya. 

We have also included the purpose in the background of the study and summarized text to reflect the gap and the scope of the study. 

3. The authors should give a justification for the study design used. 

Response: A qualitative exploratory study was conducted between December 2019 to May 2020 involving in-depth interviews to explore the views of healthcare providers to obtain an in-depth understanding of the perceived leadership roles of clinical leaders that may influence patient outcomes and health system improvements. 

4. It would be nice to have a detailed description of the study participants and how they were sampled, in the relevant section. This is for the readers, not the reviewer.

Response: We have separated the sections in the methods to highlight: study design, study setting, participant sampling and recruitment, data collection procedures, data management and analysis to provide clarity and coherence in the description 

In the ‘participant sampling and recruitment section’ we described as follows;

“The participants in our study involved frontline healthcare providers (clinicians, nurses, counselors, and a pharmacist). The total population of healthcare providers at MTRH-AMPATH was 50 at the time of the study. We first presented the research permits for conducting the study in AMPATH to the clinical leaders in charge of various clinics to explain the purpose of the study and obtain approval to interview the HCPs. We purposively and conveniently approached 25 HCPs individually and 22 consented to participate in in-depth interviews. Literature suggests that a sample size of 10 is adequate if a homogenous group of people is interviewed, nevertheless, a sample size of 15-30 is considered appropriate for a qualitative study (22)”. 

5. The data analysis section is unconvincing. First, the authors state that data was managed in Nvivo software. Which version of Nvivo? And what exactly was done using Nvivo? Then what is subsequently stated doesn’t seem to suggest that there was any computer-assisted data analysis. It appears as if the coding and analysis was done manually, but the authors feel obliged to mention Nvivo. The authors need to tell what exactly was done for analysis. Also note that you can’t increase the quality of the data by just counterchecking codes. The quality of the data is determined at the point of data collection. 

Response: “Thematic analysis was used to analyze qualitative data (23,25). The first author began by transcribing the audio recordings verbatim then the transcripts were read and re-read by all the authors to familiarize themselves with the content of the data and understand its richness and diversity. The transcribed data was imported to NVivo12 for management and coding. Two authors with qualitative backgrounds coded the data, compared, discussed, and examined the codes and categories to identify patterns and themes for interpretation. All the authors then read and discussed the themes and agreed on the final themes to be included in the report. A final codebook indicating emerging themes was then developed that informed the final write-up of the results and supported by relevant quotes from the data. To ensure validity and reliability, and increase the quality of the data, the codes were counterchecked and appraised by all authors to confirm there was no repeated and conflicting content (26,27). The data collection procedures and analysis were made clear to help maintain consistency and rigor throughout the study. Team debrief sessions were held during the data collection to enhance the quality of the data. The analysis informed descriptive categories/domains that represent thematic healthcare providers' perceptions of the clinical leadership roles of clinical leaders in an HIV primary care facility”.

6. The manuscript still requires some language editing. 

Response: We have reviewed the entire manuscript and made necessary edits to ensure coherence in language and general flow

---

## [Decision Letter · Decision Letter 2]

20 Feb 2024

PONE-D-23-05699R2Leading from the Bottom: The Clinical Leaders Roles in an HIV Primary Care Facility in Eldoret, KenyaPLOS ONE

Dear Dr. Cherop,

Thank you for submitting your manuscript to PLOS ONE. After careful consideration, we feel that it has merit but does not fully meet PLOS ONE’s publication criteria as it currently stands. Therefore, we invite you to submit a revised version of the manuscript that addresses the points raised during the review process.

We look forward to receiving your revised manuscript.

Kind regards,

Edward Nicol, PhD

Academic Editor

PLOS ONE

Journal Requirements:

Reviewers' comments:

Reviewer's Responses to Questions

**Comments to the Author**

1. If the authors have adequately addressed your comments raised in a previous round of review and you feel that this manuscript is now acceptable for publication, you may indicate that here to bypass the “Comments to the Author” section, enter your conflict of interest statement in the “Confidential to Editor” section, and submit your "Accept" recommendation.

Reviewer #1: All comments have been addressed

Reviewer #2: All comments have been addressed

2. Is the manuscript technically sound, and do the data support the conclusions?

Reviewer #1: Yes

Reviewer #2: Partly

3. Has the statistical analysis been performed appropriately and rigorously? 

Reviewer #1: N/A

Reviewer #2: Yes

4. Have the authors made all data underlying the findings in their manuscript fully available?

Reviewer #1: No

Reviewer #2: Yes

5. Is the manuscript presented in an intelligible fashion and written in standard English?

Reviewer #1: Yes

Reviewer #2: Yes

6. Review Comments to the Author

Reviewer #1: No further comments. The authors have fully addressed all concerns earlier raised; and i am happy with the submission.

Reviewer #2: The distinctions in the categories between Middle- Level Leadership and Supervisory roles is not always clear. Perhaps it would be better to organize the themes around functions, such as "establishing linkages between health system levels," supervision, etc.

7. PLOS authors have the option to publish the peer review history of their article (what does this mean?). If published, this will include your full peer review and any attached files.

Reviewer #1: No

Reviewer #2: No

---

## [Author Response · Author response to Decision Letter 2]

3 Mar 2024

Journal Requirements:

Response: The reference list has been reviewed to ensure it is updated. 1 journal article article of “Oates K. The new clinical leader. Vol. 48, Journal of Paediatrics and Child Health. 2012. p. 472–5.: / which was initially cited in APA format has now been cited in the current citation format and included as citation No.37 in the body document and reflected in the reference list. Retracted papers are not applicable in the references used in this document. 

6. Review Comments to the Author

Reviewer #1: No further comments. The authors have fully addressed all concerns earlier raised; and i am happy with the submission.

Reviewer #2: The distinctions in the categories between Middle- Level Leadership and Supervisory roles are not always clear. Perhaps it would be better to organize the themes around functions, such as "establishing linkages between health system levels," supervision, etc.

• I have reorganized this section to provide a sub-title “ Interconnecting health systems and supervisory oversight roles” that would describe the themes around the different leadership levels of the health systems. I have provided a brief introduction of the section to reflect the interconnectedness and the supervisory oversight roles of the clinical leader in the health system. 

• This is the brief introduction “Participants reported that clinical leaders play an important role in interconnecting different levels of health systems management by facilitating communications, collaboration, and coordination among various stakeholders. Through effective supervisory oversight, the leaders ensure the delivery of high-quality HIV care while fostering professional growth and development within their teams. 

• The emerging themes and quotes around interconnectedness and supervisory oversight have been reorganized to fit the 2 sub-themes. 

• The reorganization of the themes has been reflected in the results section, the discussion section to enhance a good flow of the themes, and the sub-title revised in the abstract to reflect 3 roles of a clinical leader.

---

## [Editor Report · Decision Letter 3]

27 Mar 2024

Leading from the Bottom: The Clinical Leaders Roles in an HIV Primary Care Facility in Eldoret, Kenya

PONE-D-23-05699R3

Dear Dr. Cherop,

We’re pleased to inform you that your manuscript has been judged scientifically suitable for publication and will be formally accepted for publication once it meets all outstanding technical requirements.

Kind regards,

Edward Nicol, PhD

Academic Editor

PLOS ONE